## [Peer Review File · Nature Methods]

Next-generation multicolor indicators for in vivo imaging of norepinephrine

Corresponding Author: Professor Tommaso Patriarchi

Version 0:

Decision Letter:

14th May 2025

Dear Tommaso,

Let me first sincerely apologize for the delay in providing you with a decision. Your Article, "Next-generation multicolor indicators for in vivo imaging of norepinephrine", has now been seen by three reviewers. As you will see from their comments below, although the reviewers find your work of considerable potential interest, they have raised a number of concerns. We are interested in the possibility of publishing your paper in Nature Methods, but would like to consider your response to these concerns before we reach a final decision on publication.

We therefore invite you to revise your manuscript to address these concerns. Given the feedback from the reviewers, we do however recommend shifting the focus of the manuscript towards the green sensor.

Link Redacted

We hope to receive your revised paper within 2-3 months. If you cannot send it within this time, please let us know. In this event, we will still be happy to reconsider your paper at a later date so long as nothing similar has been accepted for publication at Nature Methods or published elsewhere.

OPEN SCIENCE REQUIREMENTS

REPORTING SUMMARY AND EDITORIAL POLICY CHECKLISTS

EXTENDED DATA FIGURES

DATA AVAILABILITY

All novel DNA and RNA sequencing data, protein sequences, genetic polymorphisms, linked genotype and phenotype data, gene expression data, macromolecular structures, and proteomics data must be deposited in a publicly accessible database, and accession codes and associated hyperlinks must be provided in the "Data Availability" section.

MATERIALS AVAILABILITY

As a condition of publication in Nature Methods, authors are required to make unique materials promptly available to others

without undue qualifications.

ORCID

Please do not hesitate to contact me if you have any questions or would like to discuss these revisions further. We look forward to seeing the revised manuscript and thank you for the opportunity to consider your work. Again, I am very sorry about the delays.

Best regards,
Nina

Nina Vogt, PhD
Senior Editor
Nature Methods

Reviewers' Comments:

Reviewer #1 (Remarks to the Author):

The manuscript by Rohner et al. reports the development and validation of a new generation of blue-shifted and red-shifted norepinephrine indicators, termed nLightG2 and nLightR2, respectively. The authors first conducted combinatorial screening of indicator mutations to identify potential mutations that resulted in improved performance. Through a series of methodical validations and characterizations in *ex vivo*, *in vitro*, and *in vivo* settings, and finally in awake behaving animals, the authors provided solid evidence demonstrating the enhanced brightness and improved dynamic range of nLightG2 and nLightR2 compared to their previous generation.

Overall, I believe the development of the novel nLightG2 and nLightR2 indicators is timely and highly beneficial to the community. I am particularly excited by the red-shifted nLightR2, as it will allow for simultaneous dual-color imaging of the dynamics of two different neurotransmitters using currently available green neurotransmitter indicators. That being said, I believe the authors should provide additional characterizations and further developments related to nLightR2 to increase the impact of these NE indicators.

Major:

1. Their data showed that nLightR2 emitted green fluorescence under blue excitation, suggesting crosstalk between green and red channels when using dual-color imaging setups involving blue excitation (lines 423–428). Although I appreciated the authors' transparency in reporting this imperfection of nLightR2, my enthusiasm was dampened because this limitation restricts its use in dual-color fiber photometry experiments. While the authors mitigated this limitation by employing a spatial segregation strategy to conduct dual-color imaging of NE dynamics in CA1 neurons (nLightR2) and calcium activity in CA1 astrocytes (GCaMP6f), this approach requires two-photon microscopy. Is it possible for the authors to use other red fluorescent proteins instead of the mApple fluorophore in nLightR2? The authors suggested using a combination of red-light-excited opsins with the green NE indicator nLightG2 (lines 492–493). However, it is unclear whether current red-shifted calcium indicators exhibit similar photoswitching properties as nLightR2. At a minimum, the authors should demonstrate the feasibility of using nLightG2 with some red-shifted calcium indicators for dual-color imaging of the same neurons.
2. Related to point 1, given this limitation, I am confused about how to interpret the results in Figure 4, where dual-color fiber photometry was used to simultaneously image neuronal activity via GCaMP8f and NE dynamics via nLightR2 in the LC during sleep.
3. The authors provided conclusive evidences showing that nLightR2 and nLightG2 have a larger dynamic range than nLightR and nLightG at μM NE levels. However, the NE concentration in the extracellular space of the brain is at nM levels. It would greatly help future users to understand what to expect if the authors could provide a side-by-side comparison of nLightR2/G2 with GRAB_NE and nLightR/G in response to physiologically relevant NE levels (i.e., 0.1–10 nM).
4. I wonder why the authors did not examine the recruitment of miniG proteins for nLightG2.

5. Figure 1j–l: Why was the kinetics of nLightG2 not measured here?
6. Figure 2e & 2l: Related to the point above, it appears that nLightR2 has faster kinetics. Additionally, in the same stimulation setup, nLightR2 has a peak $\Delta F/F_0$ of 1.5%, while nLightG2 has a peak $\Delta F/F_0$ of 50%. Does this suggest that nLightG2 is approximately 30-fold brighter than nLightR2, which is inconsistent with the results in Figure 1e,f? Or do nLightG2 and nLightR2 respond differently at low NE levels (i.e., nM) compared to μ M ranges?
7. Figure 3: The authors used only nLightR2 to measure the NE response to otogenetic LC stimulation. I think this is a good opportunity for the authors to conduct a side-by-side comparison of the peak $\Delta F/F_0$ of nLightR2 and nLightG2 in response to physiologically relevant NE levels.
8. The analyses of fiber photometry data appeared to differ across the three experiments. For example, some used isosbestic correction, while others did not. I think it would help the audience interpret and compare the results across these experiments if the analyses were consistent.

Minor:

1. Figure 1, the concentration of NE and Trz is 10 nM in panel e, and 10 M in panel f and h. But there are 10 μ m in the main text (line 147-148).
2. Figure 5p: Why did NE signals start to decrease ~1.5–2 seconds before the rewards? If the animal runs at a speed of 20 cm/s (panel q), this means it was ~30–40 cm away from the reward location, which is relatively far given that the reward location is 85 cm from the start.
3. Line 202 co expressed -> co-expressed
4. Line 222: typo: desing -> design
5. Line 320 “The analysis revealed ...” what analysis? ANOVA? How to interpret the interaction between the indicator type and the task phase?
6. Line 351: what test did you use for figure 4i?
7. Line 602: 3.33·10⁶ -> 3.33×10⁶
8. Line 620, why only photobleaching was tested for nLightR2?
9. Line 657: 0.5·10⁶ -> 0.5×10⁶
10. Line 658: 20fold -> 20-fold
11. Line 927 and 930: “Mann-Whitney U test was performed ...”: Normality tests should be conducted first to determine whether a t-test or Mann-Whitney U test will be performed.

Reviewer #2 (Remarks to the Author):

The study by Rohner et al. presents the development and validation of two new genetically encoded norepinephrine (NE) sensors: green nLightG2 and red nLightR2. These second-generation indicators demonstrate improved $\Delta F/F$ responses and higher affinity compared to their first-generation counterparts previously reported by the same group. The authors adopt a similar experimental pipeline to their earlier publication—characterizing sensor performance in HEK293 cells, primary cultured neurons, brain slices, and multiple in vivo behavioral paradigms—with particular focus on the red-shifted sensor, nLightR2.

While the manuscript offers solid evidence that the new sensors outperform their predecessors, I am less convinced by the extent of improvement, especially for nLightR2. Despite optimization, nLightR2 still underperforms and, in my view, does not yet represent a viable tool for widespread use. Although the need for a red NE sensor to complement green calcium indicators like GCaMP is clear, the current data do not support nLightR2 as a practical solution. I would encourage the authors to shift focus to nLightG2, which shows substantially better performance, and to explore its compatibility with established red calcium indicators such as RCaMP3 or XCaMP-R—both of which approach the performance level of GCaMP5/6.

A key need in the field is a sensitive NE sensor—green or red—that can reliably detect spontaneous, physiologically relevant NE fluctuations, akin to what the latest dopamine sensors achieve. nLightR2 clearly falls short of this benchmark. I would strongly recommend the authors use nLightG2 in the experiments shown in Figures 4–6 to convincingly demonstrate that it is currently the best available NE sensor. An ideal addition would be a direct in vivo comparison between nLightG2 and GRABNE2m.

Specific Comments:

1. Figure 2: nLightR2 exhibits minimal difference in response between 20 nA and 200 nA iontophoretic injections, whereas nLightG2 displays a nearly tenfold increase under the same conditions. Given the similar EC₅₀ values, this disparity likely reflects the poor dynamic range of nLightR2, which is further supported by its weak response to electrical stimulation (<0.015 $\Delta F/F$).
2. Figure 4: The in vivo performance of nLightR2 appears questionable. In panel d, the correlation between nLightR2 and GCaMP is weak, assuming this trace represents their best example. Furthermore, cross-correlation analyses in panels e and g are performed around selected peaks; such selective analysis is insufficient. To demonstrate true signal correlation, the entire recording should be analyzed. Additionally, the use of “Z value” for cross-correlation is unclear—standard cross-correlation values range from -1 to 1. Clarification is needed. The sleep-state labeling (wake/NREM/REM) in panels d and f lacks accompanying EEG/EMG traces or power spectra, and the annotations do not appear to match the observed GCaMP activity.
3. Figures 5 and 6: These experiments lack critical controls. It remains unclear whether the observed changes in nLightR2 signals truly reflect changes in NE levels. The authors should include representative, continuous dual-color recordings (similar to Fig. 4d, not just averaged traces) aligned with key behavioral events like locomotion and reward delivery. This would provide more convincing evidence of sensor performance and system reliability.
4. Manuscript Preparation: The manuscript contains numerous errors and inconsistencies, which detract from the overall quality and rigor of the work. Examples include:

Fig. 1e: concentration listed as "10 mM"; Fig. 1f and 1h: "10 M" (should be μM as per the legend).
Fig. 3 legend is misnumbered.

Extended Data Fig. 3: panel labels incorrect (two "d" panels).

Extended Data Fig. 5e: incorrect x-axis label; legend panel "d" should refer to "TL", not "TF".

These cumulative oversights suggest a lack of thorough manuscript review prior to submission.

Minor Comment: In Fig. 1f, the black and purple bars are difficult to distinguish and could benefit from improved contrast or labeling.

Reviewer #3 (Remarks to the Author):

Norepinephrine (NE) is one of the most important neuromodulators in the brain. Understanding the precise spatiotemporal dynamics of norepinephrine is essential for understanding how it alters neural circuits to change arousal states. Rohner et al. report the development of two new norepinephrine sensors, nLightG2 and nLightR2, that show significantly improved brightness and dynamic range over previous NE sensors. By employing combinatorial testing of reported modifications of similar fluorescent sensors, the authors demonstrate that their new sensors display large fluorescence ranges, 2139% and 740% $\Delta F/F_0$ for the green and red sensors, respectively. Through in vitro, ex vivo, and in vivo experiments they demonstrate how the improvement in dynamic range of the red sensor, especially, can be utilized to study NE dynamics in varying contexts.

Originality and significance

The technical novelty for this class of sensors is moderate, because it uses existing mutations from other sensors to improve dynamic range, and the photoactivation by blue light of the red sensor is still an issue. Biologically, the experiments do not demonstrate significant advances in the spatiotemporal NE dynamics that are revealed by these new sensors. However, the improved sensors show impressive quantitative improvement over existing sensors and will be valuable to the neuroscience community.

Major Comments

1. While many characterizations are done, there are a few important characterizations missing that are needed to help evaluate the sensors.

1A The characterization of nLightG2 is much less than that of nLightR2. For example, the kinetics of nLightG2 are important but not shown.

1B pH sensitivity of both sensors

1C Perturbation of in vivo physiology: how much do these exogenous overexpressed reporters buffer endogenous NE concentrations?

2. NE recordings across sleep/wake cycles (Fig. 4) are different from previous recordings.

2A Previously, NE release declines throughout NREM and REM (for example, PMID: 35798980), and shows infraslow oscillations during sleep. This is not seen in the presented data.

2B Both nLightR2 and its control reporter show tonic increases in fluorescence in both example traces and mean activity. Is this due to photoactivation by blue light during the simultaneous Gcamp recording? The drift needs to be explained and photoactivation artifact needs to be considered.

2C Based on the example trace, the correlation between nLightR2 and jGcamp8f is not high.

2D The data presented in Fig. 4 can also be better evaluated if spectrograms for EEG can be shown in the same figure.

3. The magnitude of the fluorescent sensor response shown in figure 5e are very small. The lack of results with a control sensor makes it hard to interpret such small effects.

Specific Comments

1. Given the on rate of the sensor that is 52ms, a 10Hz low pass filter may result in the lack of detection of NE peaks.

2. The concentrations shown in panels 1e, f, h, and m should be in μM .

3. In panel 1b, the final image in the merged file does not fully render except in the full-page version of the figure.

4. Figure 3 is missing panel e. The figure captions do not match the figures.

5. Extended Figure 3: is $n=3$ giving sufficient statistical power for evaluation of e.g. beta-arrestin coupling of these sensors?

6. Fig. 6d: there does not seem to be much correlation between Gcamp and nLightR2

7. Fig. 6m-o: it is unclear how this conclusion is reached based on these data: "signals extracted from one astrocytic ROI correlated with NE signals extracted from the NE ROI positioned closest to the astrocytic ROI"

Version 1:

Decision Letter:

Our ref: NMETH-A58375A

8th Oct 2025

Dear Tommaso,

Thank you for submitting your revised manuscript "Next-generation multicolor indicators for in vivo imaging of norepinephrine" (NMETH-A58375A). As I mentioned before, it has been seen by the original referees and their comments are below. The reviewers find that the paper has improved in revision, and therefore we'll be happy in principle to publish it

in Nature Methods, pending minor revisions to satisfy the referees' final requests and to comply with our editorial and formatting guidelines. Please make sure to incorporate the clarifications you provided into the manuscript.

TRANSPARENT PEER REVIEW

Nature Methods offers a transparent peer review option for new original research manuscripts. We encourage increased transparency in peer review by publishing the reviewer comments, author rebuttal letters and editorial decision letters if the authors agree. Such peer review material is made available as a supplementary peer review file. **Please state in the cover letter 'I wish to participate in transparent peer review' if you want to opt in, or 'I do not wish to participate in transparent peer review' if you don't.** Failure to state your preference will result in delays in accepting your manuscript for publication.

ORCID

Author names using non-Roman characters

Nature Portfolio journals can support presentation of author names using non-Roman characters in the HTML version of the article. If you wish to, please include author names in parentheses after the Roman-character spelling; [see example online here](https://www.nature.com/articles/s44222-024-00258-2). Currently supported scripts are: Arabic, Chinese, Cyrillic, Devanagari, Greek, Hebrew, Hangul, Japanese and Persian. You will be asked to verify the rendering is correct at proof stage.

Best regards,
Nina

Nina Vogt, PhD
Senior Editor
Nature Methods

Reviewer #1 (Remarks to the Author):

I thank the authors for their efforts in addressing my comments. They have done an excellent job revising the manuscript. I have several minor comments that I think the authors should address before the paper can be published.

1. Figure 2: For nLightR2, panel d does not appear to agree with panel c. In panel d, the dF/F response seems to plateau at 20 nA stimulation, whereas panel c shows a graded increase in dF/F with stimulation current.
2. Figure 2 f & m: It is unclear what the 3D surface plot represents. Is it the 3D version of the eStim 2D plot shown on the left?
3. Figure 3b: what's "aTH"? a typo?
4. Line 386, 504, 505, 523, 526 and several other places: the details of stat test are missing.
5. The manuscript reports p values in two different formats. e.g. line 107 and 128 vs line 138. They should be consistent throughout the manuscript.
6. *, **, or *** have been added before some p values. I don't think that is necessary.
7. Line 132: citation should be moved to the end of the sentence.
8. Both Dbh-Cre and Dbh-iCre mice were used, presumably because the experiments were done in different labs. For the Dbh-iCre mice, it is unclear when and how Cre was induced.

Reviewer #2 (Remarks to the Author):

The revised manuscript is much improved, and the authors have addressed all of my concerns. My only suggestion is to use thinner lines for the traces in the new Figure 6, which would better match the other figures and may reveal additional details.

Reviewer #3 (Remarks to the Author):

The manuscript by Rohner et al. strengthened significantly. Notably, the increased focus on nLightG2, including more thorough characterization (for example, kinetics, pH sensitivity, lack of physiological perturbation), in vivo demonstration of its capacity, and side-by-side comparison with GRAB(NE2m), provides compelling evidence that nLightG2 is a much improved sensor and will help with its adoption by the scientific community. However, the following points need to be addressed further.

The most major critique comes from the analysis of NE in sleep-wake studies with nLightR2.

- The difference of the dynamics in this manuscript from that reported by previous sensors, notably the lack of the slow and almost steady decrease throughout NREM, remains unexplained (compare Figure 4 in this manuscript with PMID: 35798980). These differences could lead future users to draw qualitatively different conclusions on NE dynamics and functions. Explanation of the discrepancy, investigation of whether the discrepancy also occurs for nLightG2 will be critical for future users to understand the power and limitations of these tools.
- In addition, there are clear baseline changes in the traces Fig. 4 & Rebuttal Figure 1 & 2 for both nLightR2 sensor and mutant controls. This exists in even short recordings, and as the authors acknowledged, occur most prominently in long-wake recordings in a manner that is independent of NE. This means the sensor is responding to some other environmental variables or photoactivation. Addressing this is critical, because it is important for future users to understand when to use and when not to use this sensor, and draw appropriate conclusions when the sensor could be detecting something other than NE.

More specific comments include the following.

- Fig. 5 (former Fig. 6): We still observe minimal correlation between nLightR2 and GCaMP during reward epochs. It is also unclear why the authors would replace the original images with new representative images to illustrate the correlation – are these images both representative or selected based on the correlation?
- Fig. 6
 - o The supplementary videos clearly display striking improvements in the ability of nLightG2 to detect fast, localized NE release. Despite this, the figure falls short in effectively illustrating this. Adding additional spatial analyses to illustrate distinct clusters of NE release during looming/locomotive behavior, and illustrating these for nLightG2 and GRAB(NE2m) side by side in the main figure would highlight the improvement of the sensor.
 - o It is unclear whether NE responds to looming or the locomotion that often associates with looming. Further temporal and cross-correlational analyses could answer this question.
- Supp. Fig 2: Our questions of the buffering effects of nLightG2 were answered in part by the additional in vitro experiments. The in vivo behavioral results, however, are difficult to interpret. In the rebuttal, the authors state that nLightG2 or its control sensor was expressed bilaterally, but in the figure it shows unilateral injection, in line with other experiments in the manuscript. It is unclear that behavior perturbation would be observed if the sensor was delivered unilaterally.
- The justification for 10Hz low pass filter for a sensor with a much faster on rate is unsatisfactory – very likely, NE peaks would be missed, and the amplitude of change is underestimated. Presumably, the authors already have raw data at higher sampling rate and could apply the appropriate frequency (at least 40Hz given the on rate) for the low pass filter. What do the data look like with such filtering? What is the rationale of not doing this?

Reviewer #4 (Remarks to the Author):

I co-reviewed this manuscript with one of the reviewers who provided the listed reports. This is part of the Nature Methods initiative to facilitate training in peer review and to provide appropriate recognition for Early Career Researchers who co-review manuscripts.

We would like to thank both the Editor and all Reviewers for their positive and constructive feedback on our manuscript. As suggested, we have reorganized the manuscript to shift the focus on nLightG2 and performed additional in vitro and in vivo experiments, as described in our point-by-point responses below, shown in blue. Cited text from the revised manuscript is shown in *blue italics*. The original Reviewer's comments are shown in black *italics*. We also highlighted all major text changes in blue directly in the revised manuscript, to enable easy identification.

Reviewers' Comments:

Reviewer #1 (Remarks to the Author):

The manuscript by Rohner et al. reports the development and validation of a new generation of blue-shifted and red-shifted norepinephrine indicators, termed nLightG2 and nLightR2, respectively. The authors first conducted combinatorial screening of indicator mutations to identify potential mutations that resulted in improved performance. Through a series of methodical validations and characterizations in ex vivo, in vitro, and in vivo settings, and finally in awake behaving animals, the authors provided solid evidence demonstrating the enhanced brightness and improved dynamic range of nLightG2 and nLightR2 compared to their previous generation.

Overall, I believe the development of the novel nLightG2 and nLightR2 indicators is timely and highly beneficial to the community. I am particularly excited by the red-shifted nLightR2, as it will allow for simultaneous dual-color imaging of the dynamics of two different neurotransmitters using currently available green neurotransmitter indicators.

We sincerely appreciate the Reviewer's recognition of our work, and the enthusiasm for the tools' potential to benefit the neuroscience community.

That being said, I believe the authors should provide additional characterizations and further developments related to nLightR2 to increase the impact of these NE indicators.

Major:

1. Their data showed that nLightR2 emitted green fluorescence under blue excitation, suggesting crosstalk between green and red channels when using dual-color imaging setups involving blue excitation (lines 423–428). Although I appreciated the authors' transparency in reporting this imperfection of nLightR2, my enthusiasm was dampened because this

limitation restricts its use in dual-color fiber photometry experiments. While the authors mitigated this limitation by employing a spatial segregation strategy to conduct dual-color imaging of NE dynamics in CA1 neurons (nLightR2) and calcium activity in CA1 astrocytes (GCaMP6f), this approach requires two-photon microscopy. Is it possible for the authors to use other red fluorescent proteins instead of the mApple fluorophore in nLightR2?

We understand the Reviewer's concerns. In the initial submission of the manuscript, we reported the presence of green-emitting (500-550 nm) spots that are visible when the nLightR2 is excited in the brain of living animals by two-photon microscopy at wavelengths close to 900 nm (Supplementary Fig. 4a).

During this revision, we conducted in vitro experiments in cultured neurons expressing either nLightR2 or no indicator (untransduced) that were imaged using two-photon microscopy (Supplementary Figure 4b). Under these conditions, we determined that the presence of green emission of unknown nature was still present regardless of whether the indicator was expressed or not (Supplementary Fig. 4c-f). We additionally saw these spots when illuminating the sample using one-photon microscopy (488 nm laser). We tested whether this one-photon excited fluorescence was sensitive to NE and found no change in intensity upon exogenous application of NE (10 μ M, Supplementary Figure 4g). These observations suggest that this spurious green fluorescence emission is likely dependent on endogenous autofluorescent molecules and not on a green species of nLightR2. As a result of these new experiments and their implications, we expanded the results section on this point by referring to the Supplementary Figure as follows:

“As an important control, we first imaged nLightR2-expressing animals at 920 nm, the wavelength used to excite fluorescence in the green channel, and observed some green fluorescence emission (Extended Data Fig. 7a-b). The green fluorescence was visible upon illumination at 920 nm, but much less at 1040 nm (Extended Data Fig. 7a-b). Moreover, upon 920 nm excitation, the green fluorescence was not homogeneously distributed across the cells but was concentrated in what appeared to be somatic puncta (Supplementary Fig. 4a-b). This observation could be compatible with the possibility that the nLightR2 indicator may partition in two species as previously shown for other red-shifted indicators based on the mApple fluorophore⁴². Alternatively, the green fluorescence observed upon 920 nm excitation could be due to endogenous autofluorescence. In line with this hypothesis, the puncta were also observed in non-transduced cultured neurons using two-photon excitation. Additionally, those puncta had broad excitation/emission spectra and showed high green-red colocalization. This was not the case for neurons expressing nLightR2 where the brightest red pixels had low

green fluorescence intensity. More importantly, these puncta did not respond to application of NE in cultured neurons (10 μ M, Supplementary Fig. 4c-g)."

As the Reviewer correctly suggests, a direction for future development of new red-shifted GPCR-based indicators would consist in adopting a different red fluorescent protein scaffold. This is certainly an area of great interest to us and we are actively pursuing this direction in our laboratory. However, considering that all currently available red GPCR-based indicators are based on cpmApple, such new developments necessarily require labor- and time-intensive large-scale screening efforts and fall beyond the scope of the current manuscript.

The authors suggested using a combination of red-light-excited opsins with the green NE indicator nLightG2 (lines 492–493). However, it is unclear whether current red-shifted calcium indicators exhibit similar photoswitching properties as nLightR2. At a minimum, the authors should demonstrate the feasibility of using nLightG2 with some red-shifted calcium indicators for dual-color imaging of the same neurons.

We thank the Reviewer for this suggestion and have included a new experiment in Figure 4 where we used nLightG2 together with the recently developed red-shifted Ca^{2+} indicator PinkyCaMP (Fink et al, BioRxiv 2024; PMID: 39763884) in the amygdala during cued-fear conditioning via fiber photometry (Figure 4j-s). We additionally transduced another cohort with PinkyCaMP and nLightG2-ctr (NE-insensitive) to be able to carefully assess potential artifacts. We found that the combination of nLightG2 with PinkyCaMP worked well for addressing behaviorally relevant changes as we were able to detect neural activity adaptations in the amygdala that occurred after fear learning (Figure 4j-s). When performing dual-color photometry, the following concerns exist so we will address them individually.

1. **Bleed-through from the green to the red detector.** Despite sinusoidal modulation (see explanation below on the next point from the Reviewer) of each individual LED for excitation, very bright – and slower than the modulation frequency – fluorescence peaks from any green-emitting GEF1, could potentially be detected in the red-channel. This was not the case in our experiment shown by the fact that the largest nLightG2 peak (after foot-shock) did not coincide with the PinkyCaMP peak. In fact, PinkyCaMP was almost at baseline when nLightG2 peaked. This was also observed on the trial-by-trial basis where we found that the foot shocks produced a sharp PinkyCaMP response and a slow and tonic increase of nLightG2 signal. A correlation analysis showed a strong positive correlation with a subsequent negative anticorrelation

(Extended Data Figure 6a,b). If there was any bleed-through, one would expect that for every nLightG2 peak, there should be a slight increase in the long-wavelength ('red') detector, and thus lead to a stronger correlation. We found only a weak correlation before/after the foot-shock in the nLightG2 + PinkyCaMP group. This weak correlation is not present in the nLightG2-ctr + PinkyCaMP group, which suggest that activity of amygdala neurons and NE local release might be slightly correlated and that it is not an artifact of the photometry detection.

2. **Identity of signals.** In this experiment we looked at the activity (CaMKii α +) of amygdala neurons together with local NE release. A cross-correlation analysis in a time window flanking the foot shocks revealed that the largest coefficient was at around 3 s, suggesting a system-level phenomenon such that activity in the amygdala leads to LC activation ultimately leading to local NE local release. In the case of nLightG2-ctr, there was a negative coefficient with little to no lag. This comes from the small dip in the nLightG2-ctr signal produced by the foot-shock, which is a non-NE related signal (as noted in the results) and is likely to be a signal contamination by hemodynamic changes (Extended Data Figure 6c,d).
3. **Hemodynamics/movement artifacts.** Many small movement- or hemodynamic-related artifacts arising during behavioral paradigms can be mitigated by isoemissive correction (e.g., 405 nm, see our recent Primer on this: Simpson E. et al, Neuron 2024; PMID: 38103545). However, strong stimuli such as electric shocks can produce residual signal contaminants that persist after correction. The use of such stimuli during photometry monitoring of neuromodulators is already broad across the literature (see for example, PMIDs: 33915107, 39009835, 37972184, 38036855), yet the careful implementation of appropriate control indicators remains only sporadic. We chose to tackle this important issue by adopting the control indicator and observed that foot-shocks lead to a small but detectible artifact in the nLightG2-ctr signal, that is clearly very different from the PinkyCaMP and the nLighG2 signals.

Finally, regarding the photoswitching of the red-emitting GEFI, we particularly chose PinkyCaMP because it is based on mScarlet and does not positively photo-switch with blue-light (465-488nm) (for the characterization and in vivo use with optogenetics of PinkyCaMP please refer to Figure 1h, and Figure 5 e,f in Fink et al, BioRxiv 2024; PMID: 39763884). The photo-switching artifact is inconvenient when blue-light is pulsed such as during optogenetic stimulation at the same site where photometry recordings occur (see particularly Taniguchi J. et al, eLife 2024, PMID: 38748470). However, this is not an issue when continuous illumination

from blue and green light is applied to co-excite green-emitting and red-emitting GEFIs during dual-color photometry recordings, as we've shown in Figure 4.

2. Related to point 1, given this limitation, I am confused about how to interpret the results in Figure 4, where dual-color fiber photometry was used to simultaneously image neuronal activity via GCaMP8f and NE dynamics via nLightR2 in the LC during sleep.

We thank the Reviewer for this very important point that is very pertinent in the field of dual-color fiber photometry. For our dual-color fiber photometry experiments, we used a two-color Doric system, which modulates the 3 different excitation LEDs (405, 475 and 560 nm) at different frequencies (respectively 208.615 Hz, 572.205 Hz and 333.786 Hz). Upon acquisition, the data is automatically demodulated around each modulated frequency, minimizing crosstalk between the different excitation frequencies. Thus, the contribution of the blue LED excitation (modulated at 572.205 Hz) to our collected signals in the nLightR2 channels (modulated at 333.786Hz) is expected to be negligible in this setting.

To nevertheless directly verify the absence of crosstalk experimentally, we recorded fiber photometry signals over the LC of a mouse co-expressing jGCaMP8f and nLightR2 while only exciting with 405 and 475 nm LEDs, both modulated at their respective frequencies. In this way, excitation of nLightR2 does not occur and any remaining signal should be due to crosstalk. In the Figure shown below, the raw signals acquired on the nLightR2 and on the jGCaMP8f emission channels are present. It is clear that - even at this raw data stage - variations in nLightR2 signal are negligible. We added the below figure in the Supplementary Information of the manuscript.

Supplementary Figure 3 | Verification of independence of nLightR2 and jGCaMP8f in frequency-modulated fiber photometry. Raw nLightR2 and jGCaMP8f signals acquired in the 460-490nm and 555-570nm emission outputs only with 405 and 475 nm LEDs excitation (excitation at 560nm is omitted).

3. The authors provided conclusive evidences showing that nLightR2 and nLightG2 have a

larger dynamic range than nLightR and nLightG at μM NE levels. However, the NE concentration in the extracellular space of the brain is at nM levels. It would greatly help future users to understand what to expect if the authors could provide a side-by-side comparison of nLightR2/G2 with GRAB_NE and nLightR/G in response to physiologically relevant NE levels (i.e., 0.1–10 nM).

We agree with the Reviewer that evaluating sensor performance at more physiologically relevant NE concentrations is important. We now performed side-by-side comparison of nLightR/R2, nLightG/G2 and GRAB_NE2m sensors upon application of 100 nM NE (shown in Supplementary Information Figure 1a,b) and added the following sentence to the manuscript:

“Next we showed that the new indicators detect low concentrations of NE more robustly by measuring the fluorescent change of nLightG, nLightG2 and GRAB_{NE2m} or nLightR and nLightR2 upon exposure to 100 nM of NE (Supplementary Information Fig. 1a,b).”

We note, however, that concentrations below 100 nM fall near or below the detection threshold for these indicators under our current experimental conditions, limiting the reliability of quantitative comparisons at sub-100 nM levels. Nonetheless, the data at 100 nM provide a meaningful assessment of sensor performance in the nanomolar range and offer practical guidance for end users. Importantly, we would like to note that even 100 nM or low micromolar NE concentrations can fall within the physiological range—particularly in proximity to synaptic release sites or during states of heightened arousal or stress, when extracellular NE levels can transiently rise well above baseline. Thus, the dynamic range observed at these concentrations remains highly relevant for in vivo applications investigating acute signaling by norepinephrine.

4. I wonder why the authors did not examine the recruitment of miniG proteins for nLightG2.

We initially chose not to perform a full characterization of nLightG2 because it was derived from a previously published scaffold (i.e., nLightG based on a sperm whale alpha-1a adrenergic receptor) with established properties described in (Kagiampaki et al, Nat Methods 2023), and we had already characterized the closely related nLightR2 in the current study. Our rationale was to avoid redundancy and focus on highlighting the novel aspects of the tool. However, we now recognize the value of including a direct characterization for completeness and to strengthen the rigor of our work; therefore, we performed new experiments to assess this. In the revised manuscript we now provide a complete signaling characterization for both nLightR2 and nLightG2. The results match nicely and indicate absence of G protein signaling for both indicators.

5. Figure 1j–l: Why was the kinetics of nLightG2 not measured here?

Please see the rationale explained above under point #4. Nonetheless, we now performed additional in vitro kinetic experiments using patch clamp fluorometry for nLightG2. The results have been included in Figure 1k,l and are described in the main text. Both indicators show comparable kinetics when exposed to 5 μ M of NE.

6. Figure 2e & 2l: Related to the point above, it appears that nLightR2 has faster kinetics. Additionally, in the same stimulation setup, nLightR2 has a peak $\Delta F/F_0$ of 1.5%, while nLightG2 has a peak $\Delta F/F_0$ of 50%. Does this suggest that nLightG2 is approximately 30-fold brighter than nLightR2, which is inconsistent with the results in Figure 1e,f? Or do nLightG2 and nLightR2 respond differently at low NE levels (i.e., nM) compared to μ M ranges?

In Supplementary Fig. 1a-b, we now provide a side-by-side characterization of the indicator response to a low physiologically relevant concentration of NE (100 nM). Under these conditions, the two indicators (nLightR2 and nLightG2) show a different maximal $\Delta F/F_0$, with the response from nLightR2 being close to 75% while that of nLightG2 approaching 200%. Overall, we do expect the indicators to show different levels of performance in brain tissue, although we would like to note that the conditions of the two experiments are very different and therefore, we would not necessarily expect the performance ratio measured in vitro to hold to the same exact extent when assessed ex vivo.

Furthermore, we apologize if the data presented in Figures 2e/l and 2a/h may have inadvertently suggested that the two indicators exhibit different kinetics, with nLightR2 appearing faster than nLightG2. This is not the case. Our new in vitro side-by-side kinetic characterization demonstrates that the two indicators possess nearly identical kinetic properties (please see the point above and the new dataset for nLightG2 in Figure 1k,l).

To avoid confusion and provide a clear message that the two multicolor indicators retain similar kinetic properties in brain tissue, we have also performed kinetic analyses on the peak-normalized signals measured ex vivo. This revealed that the average ON and OFF kinetics are also indistinguishable between the two indicator types when measured in brain slices. We have updated Figure 2 with these analyses (Figure 2o,p).

7. Figure 3: The authors used only nLightR2 to measure the NE response to optogenetic LC stimulation. I think this is a good opportunity for the authors to conduct a side-by-side

comparison of the peak $\Delta F/F_0$ of nLightR2 and nLightG2 in response to physiologically relevant NE levels.

We appreciate the Reviewer's suggestion to do a side-by-side comparison of different indicators in vivo. We focused our optogenetic experiments to the nLightR2, because the use of a red NE indicator in combination with optogenetic stimulation had never been shown before, while similar experiments had been done before for nLightG (Kagiampaki et al, Nat Methods 2023). Having said that, optogenetic stimulation is a rather artificial way of triggering NE release, as it imposes specific trains of LC stimulation that not-necessarily reproduce what is seen in a physiologically relevant condition. To therefore benchmark nLightG2 versus its predecessor nLightG as well as the other state of the art indicator GRAB_{NE2m} under more natural release conditions in vivo, we instead opted to use a well-controlled behavioral task that engages physiological NE release in the amygdala. The results from these experiments are shown in the second half of Figure 3 and clearly demonstrate the improved performance of nLightG2 over the other available indicators in detecting physiologically relevant NE levels in a behavioral context under identical conditions.

8. The analyses of fiber photometry data appeared to differ across the three experiments. For example, some used isosbestic correction, while others did not. I think it would help the audience interpret and compare the results across these experiments if the analyses were consistent.

We thank the Reviewer for the careful attention given to the analysis methods used in our photometry experiments. While we agree that consistency in data processing is generally desirable, in this case we believe that enforcing a uniform analysis pipeline across all experiments would risk obscuring, rather than clarifying, the underlying biology.

The decision to apply isosbestic correction, or other specific processing steps, was guided by the characteristics of the signal, the noise profile of the recordings, and the specific experimental objectives (considerations also explained in our recent Primer on this: Simpson E. et al, Neuron 2024; PMID: 38103545). Small differences in the analysis therefore reflect the diversity of experimental setups used in this study. For example, in the optogenetic benchmarking of nLightR2 we employed alternating 405/555 nm excitation, whereas in other experiments a sinusoidal modulation of excitation light was used. In each case, the collaborating laboratories applied the pipeline optimized for their brain region of interest and technical configuration.

We recognize that these differences can complicate direct visual comparisons across figures. To address this, we now explicitly clarify the rationale for each analysis pipeline in the revised Methods section and figure legends. We believe our approach maintains analytical rigor while ensuring that the biological relevance of each experimental condition is preserved. In addition, the use of different analysis pipelines demonstrates the robustness and the broad applicability of the sensors in different laboratory settings.

Minor:

1. Figure 1, the concentration of NE and Trz is 10 mM in panel e, and 10 M in panel f and h. But there are 10 μ m in the main text (line 147-148).

We thank the Reviewer for pointing out this discrepancy. We sincerely apologize for the confusion—this was due to a software-related rendering error in which the " μ " (mu) symbol used to denote micromolar concentrations (μ M) was not correctly recognized during figure export. As a result, it was either omitted or incorrectly displayed as "m" or an empty space, leading to misleading annotations such as "10 mM" or "10 M" in the figure panels and legends. We confirm that the correct concentration used throughout the experiments was 10 μ M for both NE and Trz, as accurately stated in the main text (lines 147–148). The figure and legend have now been corrected accordingly. We appreciate the Reviewer's attention to this detail.

2. Figure 5p: Why did NE signals start to decrease ~1.5–2 seconds before the rewards? If the animal runs at a speed of 20 cm/s (panel q), this means it was ~30–40 cm away from the reward location, which is relatively far given that the reward location is 85 cm from the start.

We thank the Reviewer for this comment. In the experiments displayed in old Fig. 5n-w and old Extended Data Fig. 9c-m, mice decrease their running speed before reaching the reward. Therefore, the observed decrease in NE signal before reward delivery might be due to mice slowing down in the portion of the virtual corridor preceding the reward location.

Please note that, following the Reviewers' suggestions, we extensively revised the new version of the manuscript to focus more on the green rather than the red indicator. Consequently, we removed the experiments presented in old Fig. 5n-v.

3. Line 202 co expressed -> co-expressed

We thank the Reviewer for catching this typo. We corrected it.

4. Line 222: typo: desing -> design

We thank the Reviewer for catching this typo. We corrected it.

5. Line 320 “The analysis revealed ...” what analysis? ANOVA? How to interpret the interaction between the indicator type and the task phase?

We thank the Reviewer for this comment. The analysis we performed was a mixed-effects ANOVA, in which indicator type was treated as a between-subject factor and task phase (Baseline, Association, Re-exposure) as a within-subject factor. The significant interaction between indicator type and task phase indicates that the effect of learning stage on NE responses depends on which indicator was used. In other words, the indicators differed in how strongly they reported CS-evoked NE changes across sessions, suggesting that they vary in their sensitivity to dynamic NE fluctuations during the progression of learning.

After performing additional comparisons with the GRAB_{NE2m} indicator as well, and now having the same sample size in all groups, we revised both the analysis and figure to reflect a more comprehensive and accurate assessment of indicator performance. Quantifying peak Z-score responses upon CS presentation, as it was performed for the footshock-evoked responses, better reflects the sensitivity and dynamic range of each indicator, as it captures the maximum evoked signal amplitude while minimizing confounds related to differences in indicator kinetics that may affect integrated measures like AUC. We now used a two-way-ANOVA and edited the manuscript text as follows:

“Two-way ANOVA was conducted on the peak z-score responses time-locked to the CS (CS-Triggered Peaks) to evaluate how indicator type and task session influenced the magnitude of NE release during learning. This analysis revealed significant main effects of the task session ($p = 0.0037$), suggesting that the stage of learning influenced NE release as anticipated in an aversive learning paradigm, and of the indicator type ($p = 0.0121$), indicating overall differences in the sensitivity among the indicators. Post hoc Tukey’s comparisons showed that nLightG2 exhibited stronger peak responses to CS presentation as compared to both nLightG and GRAB_{NE2m} across the sessions: Baseline: p (nLightG2 vs. nLightG) = 0.0460; p (nLightG2 vs. GRAB_{NE2m}) = 0.1163; p (nLightG vs. GRAB_{NE2m}) = 0.9002. Association: p (nLightG2 vs. nLightG) = 0.0129; p (nLightG2 vs. GRAB_{NE2m}) = 0.0061; p (nLightG vs. GRAB_{NE2m}) = 0.9553. Re-exposure: p (nLightG2 vs. nLightG) = 0.0318; p (nLightG2 vs. GRAB_{NE2m}) = 0.0161; p (nLightG vs. GRAB_{NE2m}) = 0.9574 (Fig. 3p).”

6. Line 351: what test did you use for figure 4i?

We thank the Reviewer for the careful attention to detail. For Figure 4i we first tested for normality and homoscedasticity and as normality was not met, we used a Mann-Whitney U test. We have now clarified this in the methods section.

7. Line 602: $3.33 \cdot 10^6$ -> 3.33×10^6

We thank the Reviewer for catching this typo. We corrected it.

8. Line 620, *why only photobleaching was tested for nLightR2?*

Extended Data Figure 4 provides a detailed characterization of the blue light-induced photoswitching behavior of nLightR2, offering end users a thorough assessment of the amplitude and kinetics of this well-documented phenomenon. Since most commonly used green indicators do not exhibit this limitation, the figure focuses exclusively on nLightR2. Panel G presents a side-by-side comparison of the mixed photobleaching and photoswitching dynamics of nLightR2 and JRGECO1a, aiming to clearly illustrate the distinction and coexistence between long-term fluorescence decay due to photobleaching and the short-term, reversible decay characteristic of negative photoswitching. For clarity, we relabeled panel G with the new subtitle “Mixed photobleaching/negative photoswitching”.

9. Line 657: $0.5 \cdot 10^6$ -> 0.5×10^6

We thank the Reviewer for catching this typo. We corrected it.

10. Line 658: *20fold* -> *20-fold*

We thank the Reviewer for catching this typo. We corrected it.

11. Line 927 and 930: *“Mann-Whitney U test was performed ...”*: *Normality tests should be conducted first to determine whether a t-test or Mann-Whitney U test will be performed.*

Please see our response to point #6 above regarding the normality test. We have now clarified this in the methods section.

Reviewer #2 (Remarks to the Author):

The study by Rohner et al. presents the development and validation of two new genetically encoded norepinephrine (NE) sensors: green nLightG2 and red nLightR2. These second-generation indicators demonstrate improved $\Delta F/F$ responses and higher affinity compared to their first-generation counterparts previously reported by the same group. The authors adopt a similar experimental pipeline to their earlier publication—characterizing sensor performance in HEK293 cells, primary cultured neurons, brain slices, and multiple in vivo behavioral paradigms—with particular focus on the red-shifted sensor, nLightR2. While the manuscript offers solid evidence that the new sensors outperform their predecessors, I am less convinced by the extent of improvement, especially for nLightR2. Despite optimization, nLightR2 still underperforms and, in my view, does not yet represent a viable tool for widespread use. Although the need for a red NE sensor to complement green calcium indicators like GCaMP is clear, the current data do not support nLightR2 as a practical solution. I would encourage the authors to shift focus to nLightG2, which shows substantially better performance, and to explore its compatibility with established red calcium indicators such as RCaMP3 or XCaMP-R—both of which approach the performance level of GCaMP5/6.

A key need in the field is a sensitive NE sensor—green or red—that can reliably detect spontaneous, physiologically relevant NE fluctuations, akin to what the latest dopamine sensors achieve. nLightR2 clearly falls short of this benchmark. I would strongly recommend the authors use nLightG2 in the experiments shown in Figures 4–6 to convincingly demonstrate that it is currently the best available NE sensor. An ideal addition would be a direct in vivo comparison between nLightG2 and GRABNE2m.

We thank the Reviewer for the insightful and constructive feedback. Based on the valuable points raised, we have decided to shift the focus of our manuscript towards the green sensor nLightG2, which demonstrates substantially stronger performance and robustness than previously available green NE sensors across experimental conditions. This focus shift allows us to present nLightG2 as the most reliable and sensitive NE sensor currently available for widespread use.

Following the suggestion, we have conducted two direct in vivo comparisons between nLightG2 and the established GRAB_{NE2m} sensor, the first using fiber photometry in the basolateral amygdala during cued-fear conditioning, and the second one using two-photon imaging in the visual cortex of awake behaving mice. Both experiments conclusively

demonstrated the improved sensitivity of nLightG2 over GRAB_{NE2m}. The in vivo two-photon data also revealed an unexpected and extremely interesting spatial microdomain organization of cortical NE release, which was not detected with either GRAB_{NE2m} or the control indicator nLightG2-ctr (please see the new Figure 6 and also the Supplementary Videos 1 and 2).

Furthermore, we performed a new dual-color photometry experiment showcasing the combination of nLightG2/nLightG2-ctr with PinkyCaMP (a new mScarlet-based red calcium sensor from Fink et al, BioRxiv 2024; PMID: 39763884). Please also see our answer to Reviewer #1.

This side-by-side analysis highlights the complementary strengths of these tools and demonstrates the suitability of nLightG2 for detecting physiologically relevant norepinephrine dynamics with high sensitivity and temporal resolution. By shifting the focus on nLightG2 and including an in vivo fiber photometry and two-photon head-to-head comparison between indicators, we provide the field with a clear, practical benchmark for NE sensor performance, while laying the groundwork for future improvements in red-shifted indicators like nLightR2.

Specific Comments:

1. Figure 2: nLightR2 exhibits minimal difference in response between 20 nA and 200 nA iontophoretic injections, whereas nLightG2 displays a nearly tenfold increase under the same conditions. Given the similar EC50 values, this disparity likely reflects the poor dynamic range of nLightR2, which is further supported by its weak response to electrical stimulation (<0.015 ΔF/F).

We would like to note that it is not unexpected that the indicators display varying sensitivities in response to a matched extracellular concentration of NE release, whether induced iontophoretically or via electrical stimulation in brain slices. As the Reviewer pointed out these differences likely reflect from the different dynamic ranges of the indicators (that is, the fluorescence ratio between NE-bound and NE-unbound states).

Please also see our answer to Reviewer 1 point #6 on this. It is important to note that this dataset is primarily intended to compare version 1.0 to version 2.0 of indicators of the same color, rather than to provide a direct comparison between the green and red variants.

2. Figure 4: The in vivo performance of nLightR2 appears questionable. In panel d, the correlation between nLightR2 and GCaMP is weak, assuming this trace represents their best example.

We acknowledge that the clarity of this figure can be improved, and we have done so in several steps. First, the new version of Figure 4 (panels d, e) presents two representative examples from many inspected examples (1,200 in total). We have also added more examples in response to comment 2C of Reviewer 3 (see below). Furthermore, corresponding EMG activity and EEG spectrograms are now added. We are confident that these improvements underscore the quality of our recordings and the usefulness of the nLightR2 biosensor recordings.

Furthermore, cross-correlation analyses in panels e and g are performed around selected peaks; such selective analysis is insufficient. To demonstrate true signal correlation, the entire recording should be analyzed.

We thank the Reviewer for bringing up this very important point. We initially had indeed planned on using entire recordings for cross-correlational analysis. However, biosensor signals need to be controlled for wake-to-sleep transitions (and associated blow flow changes) [Simpson et al., doi: 10.1016/j.neuron.2023.11.016]. Indeed, we found that both nLightR2 and nLightR2-ctr signals showed variations upon sleep-to-wake transitions that did not match the LC signal. This prompted us to restrict our analysis to NREM sleep alone. Within NREM sleep, we then focused our analysis on bouts that did not contain microarousals (MAs), again to avoid the confound of state transition-related changes in the sensors. For all these reasons, the non-MA related peaks of activity of the LC were our primary target for this analysis, as explained in the Methods.

Additionally, the use of "Z value" for cross-correlation is unclear—standard cross-correlation values range from -1 to 1. Clarification is needed.

In the revised figure, all indications should now be such that no further clarification is needed. Figure 4 panels f and g are not cross-correlation plots but rather plots of average activity of jGCaMP8f and nLightR2/nLightR2-ctr centered around jGCaMP8f-detected peaks of activity (not accompanied by MAs). This explains the use of the z-value scale bar in panels f and g. In contrast, the panel i summarizes the results from cross-correlations over all the peaks averaged in panels f and g, where the y-axis values are in the correct range for cross-correlations (between -1 and 1).

The sleep-state labeling (wake/NREM/REM) in panels d and f lacks accompanying EEG/EMG traces or power spectra, and the annotations do not appear to match the observed GCaMP activity.

The panels mentioned are now completed as requested by the Reviewer. All annotations match the observed jCaMP8f activity.

3. Figures 5 and 6: These experiments lack critical controls. It remains unclear whether the observed changes in nLightR2 signals truly reflect changes in NE levels. The authors should include representative, continuous dual-color recordings (similar to Fig. 4d, not just averaged traces) aligned with key behavioral events like locomotion and reward delivery. This would provide more convincing evidence of sensor performance and system reliability.

As requested by the Reviewer, we now provide representative and continuous nLightR2 traces aligned with key behavioral events (e.g., beginning and end of a running trial, reward delivery, new Fig. 5d).

Moreover, we also performed additional two-photon experiments in awake head-fixed mice running in the virtual corridor using the control nLightR2 sensor (nLightR2-ctr). This control sensor bears a point mutation in the binding pocket of the receptor and abolishes the response to NE, as we've shown in our in vitro characterization (Extended Data Fig. 1g). We found negligible responses of nLightR2-ctr upon running and reward delivery (Extended Data Fig. 9). nLightR2-ctr responses were compatible in amplitude with previous observations (Yogesh B., eLife 2025, PMID: 40434064). Importantly, linear models showed that the response of the nLightR2-ctr sensor did not depend on animal's speed and lick rate (Extended Data Fig. 9h-i), demonstrating that the observed nLightR2 responses (new Fig. 5 and Extended data Fig. 8) report *bona fide* NE signals.

4. Manuscript Preparation: The manuscript contains numerous errors and inconsistencies, which detract from the overall quality and rigor of the work. Examples include:

Fig. 1e: concentration listed as "10 mM"; Fig. 1f and 1h: "10 M" (should be μM as per the legend).

We thank the Reviewer for pointing out this discrepancy. As we pointed out above to Reviewer #1, we would like once again to sincerely apologize for the confusion—this was due to a software-related rendering error in which the " μ " (mu) symbol used to denote micromolar concentrations (μM) was not correctly recognized during figure export. As a result, it was either

omitted or incorrectly displayed as "m" or an empty space, leading to misleading annotations such as "10 mM" or "10 M" in the figure panels and legends. We confirm that the correct concentration used throughout the experiments was 10 μ M for both NE and Trz, as accurately stated in the main text (lines 147–148). The figure and legend have now been corrected accordingly. We appreciate the Reviewer's attention to this detail.

Fig. 3 legend is misnumbered.

We thank the Reviewer for noticing this issue. We corrected it.

Extended Data Fig. 3: panel labels incorrect (two "d" panels).

We thank the Reviewer for noticing this issue. We corrected it.

Extended Data Fig. 5e: incorrect x-axis label; legend panel "d" should refer to "TL", not "TF". These cumulative oversights suggest a lack of thorough manuscript review prior to submission. Minor Comment: In Fig. 1f, the black and purple bars are difficult to distinguish and could benefit from improved contrast or labeling.

We thank the Reviewer for noticing these minor errors. The manuscript underwent multiple rounds of internal revision and cross-checking, but as with any complex dataset, minor labeling issues can occasionally escape detection. We appreciate the Reviewer's attention to detail and have taken this opportunity to further refine the figures to ensure clarity and precision throughout the manuscript.

Reviewer #3 (Remarks to the Author):

Norepinephrine (NE) is one of the most important neuromodulators in the brain. Understanding the precise spatiotemporal dynamics of norepinephrine is essential for understanding how it alters neural circuits to change arousal states. Rohner et al. report the development of two new norepinephrine sensors, nLightG2 and nLightR2, that show significantly improved brightness and dynamic range over previous NE sensors. By employing combinatorial testing of reported modifications of similar fluorescent sensors, the authors demonstrate that their new sensors display large fluorescence ranges, 2139% and

740% dF/F0 for the green and red sensors, respectively. Through in vitro, ex vivo, and in vivo experiments they demonstrate how the improvement in dynamic range of the red sensor, especially, can be utilized to study NE dynamics in varying contexts.

Originality and significance

The technical novelty for this class of sensors is moderate, because it uses existing mutations from other sensors to improve dynamic range, and the photoactivation by blue light of the red sensor is still an issue. Biologically, the experiments do not demonstrate significant advances in the spatiotemporal NE dynamics that are revealed by these new sensors. However, the improved sensors show impressive quantitative improvement over existing sensors and will be valuable to the neuroscience community.

We sincerely thank the Reviewer for recognizing the quantitative improvements achieved with the development of nLightG2 and nLightR2. We appreciate the thoughtful evaluation and agree that the enhanced performance of these sensors will offer valuable tools for the neuroscience community.

We have conducted new in vivo two-photon imaging experiments using nLightG2 (new Figure 6) that directly demonstrate significant biological advances using our sensor. These experiments revealed localized norepinephrine signals with a resolution and sensitivity previously unattainable. Importantly, the observed signal patterns include dynamics that were neither predictable nor expected based on prior tools, underscoring that our sensor enables the detection of biologically meaningful phenomena that had remained hidden. This empirical evidence firmly establishes that the improved sensor is not only a technical advance but also a critical enabler of new biological insights.

Major Comments

1. While many characterizations are done, there are a few important characterizations missing that are needed to help evaluate the sensors.

1A The characterization of nLightG2 is much less than that of nLightR2. For example, the kinetics of nLightG2 are important but not shown.

We thank the Reviewer for highlighting this gap in our initial characterization. Following the Reviewer's recommendations, we have performed several new experiments and now provide a full characterization of nLightG2, including in vitro and ex vivo kinetic measurements (please

see Figure 1k,l and Figure 2o,p). These data show that nLightG2 displays comparable activation and deactivation kinetics to nLightR2 under matched conditions. We agree that this information is critical for fully evaluating the performance of nLightG2 and appreciate the Reviewer's suggestion.

1B pH sensitivity of both sensors

We thank the Reviewer for this important point. We have now performed a detailed pH sensitivity analysis for both nLightR2 and nLightG2. These results are included in the Supplementary Figure 1c-h. The sensors exhibit no drastic changes in their basal or saturated fluorescence, as well as in their maximal sensitivity, when sensor-expressing cells are exposed to buffers with different pH levels within the physiological range (6.0–8.0), confirming the robustness of the sensors.

1C Perturbation of in vivo physiology: how much do these exogenous overexpressed reporters buffer endogenous NE concentrations?

We thank the Reviewer for bringing up this point. Assessing potential buffering of endogenous norepinephrine (NE) signaling by genetically encoded indicators is a complex and multifactorial issue. To address this concern, we performed two complementary analyses aimed at evaluating whether overexpression of our indicators can substantially perturb physiological NE signaling.

First, we performed an in vitro assay to test whether overexpression of either one of our NE-binding sensors (nLightG2 or nLightR2) could cause a shift in the sensitivity of the wild type mouse α 1A-adrenergic receptor, by measuring a downstream signaling readout (miniGsq protein recruitment). As a negative control, we overexpressed an unrelated sensor (the orexin sensor OxLight1) that does not bind (and hence does not buffer) NE. The results, now included in Supplementary Fig. 1i, show that the NE sensitivity of the wild type receptor was not affected by co-expression of the additional NE-binding sensors, suggesting that indicator expression does not sequester NE to a degree that alters receptor activation. We added the following statement to the manuscript:

“Overexpression of GPCR-based GEFIs in vivo could potentially lead to buffering of chemical neurotransmission, as ligand molecules are sequestered by indicators near the plasma membrane, thereby reducing the ligand concentrations perceived by endogenous receptors. To investigate this, we measured the NE-induced, concentration-dependent recruitment of miniGsq proteins to the wild type A1AAR in the presence or absence of

nLightG2 or nLightR2, using the split NanoLuc complementation assay described above. No significant difference in the dose dependent miniGsq recruitment to the wild type A1AAR could be observed in the presence of nLightG2, nLightR2, or OxLight1, a GPCR-based GEF1 which does not bind NE (One-way ANOVA; $p = 0.341$; Supplementary Fig. 1i)."

Second, to probe NE buffering potential of nLightG2 in vivo, we used a cued fear conditioning paradigm, which is a behavioral assay known to be dependent on NE signaling in the amygdala, an area critically involved in fear learning. We bilaterally expressed either the functional NE indicator or the mutated, non-NE-binding version (nLightG2-ctr) in the amygdala, and used Deeplabcut for precise behavioral quantification to assess fear learning during the test. We observed no differences in either the acquisition or expression of the fear memory between the two groups, indicating that the expression of the functional sensor does not substantially interfere with NE-dependent behavioral processes in vivo. To clarify this, we added the following statement to the manuscript:

"To test whether the expression of the indicator could affect fear learning by buffering NE and interfering with its endogenous signaling, we used DeepLabCut pose estimation software to perform supervised video-based analysis of freezing behavior in animals expressing nLightG2 versus the NE binding-incompetent version of the indicator (nLightG2-ctr) in the pBLA (Supplementary Fig. 2a-c). Mixed-design ANOVA revealed a significant main effect of condition (for full post-sound epoch: $p = 1.90 \times 10^{-5}$; for CS trials: $p = 1.73 \times 10^{-8}$), but no significant effect of sensor type ($p = 0.688$) or condition \times sensor interaction ($p = 0.810$). Post hoc Wilcoxon analysis showed that freezing behavior increased significantly across sessions (Re-exposure vs Baseline: $p = 0.0117$; Re-exposure vs Association: $p = 0.0117$ for both full post-sound epoch and CS trials only), consistent with fear learning (Supplementary Fig. 2d-e). These results suggest that the observed behavioral changes were driven by conditioning and not influenced by sensor expression."

Taken together, these findings argue against significant buffering effects of the indicators and support the conclusion that our indicators can report NE dynamics without substantially perturbing endogenous NE-dependent signaling pathways.

2. NE recordings across sleep/wake cycles (Fig. 4) are different from previous recordings.

2A *Previously, NE release declines throughout NREM and REM (for example, PMID: 35798980), and shows infraslow oscillations during sleep. This is not seen in the presented data.*

2B *Both nLightR2 and its control reporter show tonic increases in fluorescence in both example traces and mean activity. Is this due to photoactivation by blue light during the*

simultaneous Gcamp recording? The drift needs to be explained and photoactivation artifact needs to be considered.

2C Based on the example trace, the correlation between nLightR2 and jGcamp8f is not high.

2D The data presented in Fig. 4 can also be better evaluated if spectrograms for EEG can be shown in the same figure.

We thank the Reviewer for these very crucial points that we clarified in the revised figure. Specifically

2A) Indeed, the LC activity does fluctuate on an infraslow timescale as in the article provided, and we hope that the new representative examples (panels, d, e) we produced make that amply clear. For further examples, please also see our response to point 2C below.

2B) The drifts in nLightR2 and nLightR2-ctr signals are mostly related to the proximity of our chosen window to a long wake episode, which increases the baseline levels of the sensors (independent of NE). We have now chosen examples of NREMS bouts that are not preceded by very long wake periods to minimize these drifts.

2C) We have modified the figure to select representative example traces containing a continuous NREMS for hundreds of seconds for both nLightR2 and nLightR2-ctrl. These examples were selected from a pool of 1,200 visually inspected samples. Each of these contain numerous LC activity surges that are highly correlated with the nLightR2 signal increase, but not with the nLightR2-ctrl signal.

We add here four more examples each for recordings with nLightR2 and nLightR2-ctrl to further illustrate the consistent differences between the two types of signals:

2024-11-28 TMcont1 snap13

2024-08-09 TM10 snap8

2024-08-08 TM05 snap10

Rebuttal-Figure 1: Additional representative examples for recordings of nLightR2-ctr injected mice. From top to bottom: vertical bar representing the vigilance state in color-code (NREMS and wake), spectrogram of EEG activity, EMG activity, and nLightR2-ctr and jGCaMP8f signals.

2024-07-22 TM08 snap 0

2024-08-14 TM08 snap 0

Rebuttal-Figure 2: Same as Rebuttal-Figure 1 for nLightR2 injected mice.

2D) We have updated our representative examples to better display our data, including, as the Reviewer suggested, spectrograms of the EEG signal.

3. *The magnitude of the fluorescent sensor response shown in figure 5e are very small. The lack of results with a control sensor makes it hard to interpret such small effects.*

Following the Reviewer's comment, we performed new two-photon experiments in awake head-fixed mice running in the virtual corridor using a control nLightR2 sensor (nLightR2-ctr). This control sensor bears a point mutation in the binding pocket of the receptor and abolishes the response to NE, as we've shown in our in vitro characterization (Extended Data Fig. 1g). We found negligible responses of nLightR2-ctr upon running and reward delivery (Extended Data Fig. 9). nLightR2-ctr responses were compatible in amplitude with previous observations (Yogesh B., eLife 2025, PMID: 40434064). Importantly, linear models showed that the response of the nLightR2-ctr sensor did not depend on animal's speed and lick rate (Extended

Data Fig. 9h-i), demonstrating that the observed nLightR2 responses (new Fig. 5 and Extended data Fig. 8) report *bona fide* NE signals.

Specific Comments

1. *Given the on rate of the sensor that is 52ms, a 10Hz low pass filter may result in the lack of detection of NE peaks.*

We thank the Reviewer for raising this point. While it is true that a 10 Hz low-pass filter imposes a temporal constraint, we believe it is compatible with the sensor's kinetics and the physiological characteristics of NE signaling in vivo. The on-rate of ~52 ms corresponds to a bandwidth of ~19 Hz, meaning the sensor is intrinsically limited in resolving features faster than this time scale. Applying a 10 Hz low-pass filter does not significantly distort the sensor's ability to capture relevant NE transients but rather serves to reduce high-frequency noise and improve the signal-to-noise ratio, particularly under in vivo imaging conditions where photon counts and motion artifacts can be limiting. Moreover, NE transients in vivo, unlike synaptic glutamate or calcium spikes, tend to occur on slower timescales (hundreds of milliseconds to seconds), particularly in extracellular space where diffusion and reuptake kinetics dominate. Therefore, we believe that a 10 Hz cutoff is sufficient to preserve the fidelity of NE dynamics relevant to the biology under investigation.

2. *The concentrations shown in panels 1e, f, h, and m should be in μM .*

We thank the Reviewer for pointing out this discrepancy. As we pointed out above to Reviewer #1, we would like once again to sincerely apologize for the confusion—this was due to a software-related rendering error in which the " μ " (mu) symbol used to denote micromolar concentrations (μM) was not correctly recognized during figure export. As a result, it was either omitted or incorrectly displayed as "m" or an empty space, leading to misleading annotations such as "10 mM" or "10 M" in the figure panels and legends. We confirm that the correct concentration used throughout the experiments was 10 μM for both NE and Trz, as accurately stated in the main text (lines 147–148). The figure and legend have now been corrected accordingly. We appreciate the Reviewer's attention to this detail.

3. *In panel 1b, the final image in the merged file does not fully render except in the full-page version of the figure.*

We thank the Reviewer for noticing this issue. We corrected it.

4. *Figure 3 is missing panel e. The figure captions do not match the figures.*

We thank the Reviewer for noticing this issue. We corrected it.

5. *Extended Figure 3: is $n=3$ giving sufficient statistical power for evaluation of e.g. beta-arrestin coupling of these sensors?*

We thank the Reviewer for this insightful comment. Given that the wild-type receptor originally used as a reference exhibits only minimal β -arrestin2 recruitment, we selected the GLP-1 receptor for a better reference signal based on our prior experience (Duffet et al., eLife, 2023). We repeated the β -arrestin2 recruitment assays using this receptor and confirmed that GLP-1 stimulation robustly recruits β -arrestin2, while NE (10 μ M) stimulation of nLightR2 and nLightG2 results in markedly attenuated recruitment. These findings support the conclusion that our indicators do not substantially engage intracellular signaling pathways.

6. *Fig. 6d: there does not seem to be much correlation between Gcamp and nLightR2*

We thank the Reviewer for pointing our attention to this point. We now provide representative and continuous nLightR2 traces aligned with key behavioral events (e.g., beginning and end of a running trial, reward delivery) so that the level of correlation in the traces can be better compared to the event triggered plots shown in new Fig. 5e-o (corresponding to old Fig. 6e-o).

7. *Fig. 6m-o: it is unclear how this conclusion is reached based on these data: "signals extracted from one astrocytic ROI correlated with NE signals extracted from the NE ROI positioned closest to the astrocytic ROI"*

The complete sentence indicated by the Reviewer reads: "Upon crossing the reward position, we found that the GCaMP6f signals extracted from one astrocytic ROI correlated with NE signals extracted from the NE ROI positioned closest to the astrocytic ROI (Figure 6m-n), while this was not the case for when the mouse started to run (Figure 6m-n)."

This sentence describes the dependence of the maximum (new Fig. 5m, corresponding to old Fig. 6m) and mean (new Fig. 5n, corresponding to old Fig. 6n) $\Delta F/F_0$ of GCaMP6f signal of a given astrocytic ROI as a function of the maximum (new Fig. 5m) and mean (new Fig. 5n) $\Delta F/F_0$ of nLightR2 signal in the nLightR2 ROI nearest to the considered astrocytic ROI. Upon crossing the reward position, we found that the linear fit of the data showed a significant

positive correlation between the considered variables (green line and asterisks in new Fig. 5m-n, $p = 0.030$ and $p = 0.025$ for maximum and mean $\Delta F/F_0$ values, respectively, permutation test with 10^4 repetitions). In contrast, when the mouse started to run we found that the linear fit of the data did not show a significant positive correlation between the considered variables (gold line and n.s. in new Fig. 5m-n, $p = 0.326$ and $p = 0.144$ for maximum and mean $\Delta F/F_0$ values, permutation test with 10^4 repetitions).

Please find our proposed changes or answers in response to the Reviewer's comments in blue, while the Reviewers' comments are shown in black italics.

Reviewer #1:

Remarks to the Author:

I thank the authors for their efforts in addressing my comments. They have done an excellent job revising the manuscript. I have several minor comments that I think the authors should address before the paper can be published.

We would like to sincerely thank the Reviewer for the appreciation of our efforts.

1. Figure 2: For nLightR2, panel d does not appear to agree with panel c. In panel d, the dF/F response seems to plateau at 20 nA stimulation, whereas panel c shows a graded increase in dF/F with stimulation current.

The Reviewer is correct that peak $\Delta F/F_0$ for nLightR2 plateaus at 20 nA in panel d, similar to the trend shown in panel b. The graded increase which is shown in panel c (3D surface plots) more closely reflects the change in the spread of the measured nLightR2 signals, rather than peak $\Delta F/F_0$. We have shown this parameter (standard deviation of fitted gaussian model) plotted against iontophoretic ejection current below.

2. Figure 2 f & m: It is unclear what the 3D surface plot represents. Is it the 3D version of the eStim 2D plot shown on the left?

Yes, the 3D surface plot in panels f and m represent the stimulated (eStim - bottom) 2D plot shown on the left.

3. *Figure 3b: what's "aTH"? a typo?*

aTH stands for anti-TH, in other words immunolabeling against the Tyrosine Hydroxylase antigen.

4. *Line 386, 504, 505, 523, 526 and several other places: the details of stat test are missing.*

We thank the Reviewer for noticing this. We corrected this.

5. *The manuscript reports p values in two different formats. e.g. line 107 and 128 vs line 138. They should be consistent throughout the manuscript.*

We thank the Reviewer for noticing this. We corrected this.

6. **, **, or *** have been added before some p values. I don't think that is necessary.*

We thank the Reviewer for noticing this. We corrected this.

7. *Line 132: citation should be moved to the end of the sentence.*

We thank the Reviewer for noticing this. We corrected this.

8. *Both Dbh-Cre and Dbh-iCre mice were used, presumably because the experiments were done in different labs. For the Dbh-iCre mice, it is unclear when and how Cre was induced.*

The term 'iCre' stands for improved Cre recombinase (i.e. a codon-improved version of the enzyme), not inducible Cre recombinase. Please see: PMID: 11835670

Reviewer #2:

Remarks to the Author:

The revised manuscript is much improved, and the authors have addressed all of my concerns. My only suggestion is to use thinner lines for the traces in the new Figure 6, which would better match the other figures and may reveal additional details.

We would like to sincerely thank the Reviewer for the appreciation of our efforts.

Reviewer #3

The most major critique comes from the analysis of NE in sleep-wake studies with nLightR2.

- The difference of the dynamics in this manuscript from that reported by previous sensors, notably the lack of the slow and almost steady decrease throughout NREM, remains unexplained (compare Figure 4 in this manuscript with PMID: 35798980). These differences could lead future users to draw qualitatively different conclusions on NE dynamics and functions. Explanation of the discrepancy, investigation of whether the discrepancy also occurs for nLightG2 will be critical for future users to understand the power and limitations of these tools.*

We would like to explain to the Reviewer that our data in Figure 4 were obtained from recordings in locus coeruleus. In contrast, the data mentioned from the previous sensors were measured in prefrontal cortex. Release and uptake dynamics are brain area-specific. Therefore, the observed differences are likely attributable to regionally specific norepinephrine signaling patterns, rather than to differences in sensor performance. Any such region specificity would need to be examined in future experiments that would need to quantify local cellular mechanisms of NE turnover.

- In addition, there are clear baseline changes in the traces Fig. 4 & Rebuttal Figure 1 & 2 for both nLightR2 sensor and mutant controls. This exists in even short recordings, and as the authors acknowledged, occur most prominently in long-wake recordings in a manner that is independent of NE. This means the sensor is responding to some other environmental variables or photoactivation. Addressing this is critical, because it is important for future users to understand when to use and when not to use this sensor, and draw appropriate conclusions when the sensor could be detecting something other than NE.*

We would like to explain to the Reviewer that we present here, for the first time, dual color fiber photometry signals from within the locus coeruleus. That is, we measured both, locus coeruleus neuronal activity and the local NE release, at the same time. Such parallel measures have never been obtained before, and they will be essential for advancing our understanding of the autoregulatory mechanisms within the locus coeruleus.

At the same time, such dual measures necessitate separate baseline calculations for both fiber photometry signals. We specify these calculations in detail in the Methods. They are done according to published procedures that we cite and they are currently the only ones available for dual color measures. They have the minor tradeoff that the nLightR2 signal is less faithfully taken into account than the GCaMP signal, in particular over the long (hundreds of seconds) recordings.

The resulting slow baseline changes are not related to ‘environmental variables’ or ‘photoactivation’. In support of this, we note that the analyses we present include all NE transients throughout the recording, yielding clear and significant differences between the nLightR2 sensor and its control.

More specific comments include the following.

• Fig. 5 (former Fig. 6): We still observe minimal correlation between nLightR2 and GCaMP during reward epochs. It is also unclear why the authors would replace the original images with new representative images to illustrate the correlation – are these images both representative or selected based on the correlation?

We thank the Reviewer for this comment. We appreciate the opportunity to clarify our choice of the traces in Figure 5d and the observed correlation values between GCaMP6f and nLightR2 during reward epochs.

The GCaMP6f traces displayed in Fig. 5d are taken from 3 astrocytic ROIs, corresponding to two cellular processes (ROI#1 and ROI#3) and to one cell body (ROI#2). The nLightR2 traces displayed in Fig. 5d are instead taken from three square ROIs close to the abovementioned astrocytic ROIs. Astrocytic ROI#1 shows signal dynamics which correlate with the signal of all three nLightR2 ROIs. It is interesting to note that signal correlation for ROI#1 is higher during the 3rd and 4th reward events occurring at around 120 s and around 160 s, respectively, and lower for the 1st and 2nd reward events. Astrocytic ROI#3 shows smaller levels of signal correlation with nLightR2 ROIs. Astrocytic ROI#2, instead, shows low frequency GCaMP6f responses. Our choice of ROIs and traces in Fig. 5d was therefore dictated by the wish to display ROI response heterogeneity linked to both the reward event (not all reward events display similar correlation between GCaMP6f and nLightR2 signals) and to the anatomical location of the ROI (e.g., process vs soma). Finally, we would like to underline that mean values of the average correlation between GCaMP6f signal and nLightR2 signals during reward epochs are reported in Fig. 5o and are calculated on 4169 ± 3362 (mean \pm SD) pairs of ROIs sampled across 7

recording sessions in 4 animals. During reward epochs average correlation is between 0.4 and 0.5.

• *Fig. 6*

o The supplementary videos clearly display striking improvements in the ability of nLightG2 to detect fast, localized NE release. Despite this, the figure falls short in effectively illustrating this. Adding additional spatial analyses to illustrate distinct clusters of NE release during looming/locomotive behavior, and illustrating these for nLightG2 and GRAB(NE2m) side by side in the main figure would highlight the improvement of the sensor.

We thank the reviewer for this thoughtful suggestion and for highlighting that the current Fig. 6 does not fully convey the striking spatiotemporal heterogeneity of the nLightG2 signal apparent in the supplementary videos. We revised the main figure and added targeted panels to better illustrate the features noted.

To illustrate sensor performance side by side, we present matched example Field of Views for nLightG2, and GRAB_{NE2m} processed identically (same tiling, baseline and color scales; Figure 6d-e) to make the microdomain-like heterogeneity and sensitivity difference between the two sensors clearer. In line with the reviewer's suggestion to further dissect the behavioral determinants of NE dynamics, we implemented a generalized linear model (GLM) to quantitatively separate the contributions of sensory (looming) and behavioral (running) variables in nLightG2 mice. Specifically, the model includes separate terms for looming stimuli, running speed, and their interaction, allowing us to partition the total fluorescence variance into components uniquely explained by each factor while accounting for shared covariance (Fig. 6l-m). This approach provides a quantitative framework to disentangle visually evoked and movement-related components of the NE signal on a per-tile basis and animal basis. The GLM revealed that the full model explained 4.4 ± 2.3 % (mean \pm SEM, N = 7 mice) of the total fluorescence variance. Removing individual predictors demonstrated that running accounted for the largest proportion of explained variance (68.0 ± 5.7 % of ΔR^2), followed by looming (20.7 ± 3.5 %) and their interaction (10.5 ± 2.8 %; Friedman $p = 9.1 \times 10^{-4}$; Wilcoxon $p = 0.016$). These results have been incorporated in the text and indicate that behavioral state exerts a dominant influence on NE activity but that visually driven components remain significant, even when movement-related variance is accounted for.

o It is unclear whether NE responds to looming or the locomotion that often associates with looming. Further temporal and cross-correlational analyses could answer this question.

Regarding the question of whether the NE signal reflects the visual loom itself or the locomotion that often follows, we note that panel 6h already demonstrates that locomotion (in a forced running paradigm) increases NE levels.

We thank the reviewer for raising this important point and for emphasizing the challenge of disentangling sensory-evoked from movement-related components of NE activity. To address this confound directly, we implemented a generalized linear model (GLM) to

quantify the relative contributions of sensory (looming) and behavioral (running) variables to NE dynamics, in line with recent studies applying this approach to dissociate sensory and motor influences on activity. As mentioned above, the GLM provides a quantitative approach for separating visually evoked from movement-related components by partitioning the explained variance (ΔR^2) attributable to each factor while accounting for shared covariation. The GLM analysis revealed that behavioral state exerts a strong influence on NE activity in nLightG2 animals, yet salient visual stimuli also contribute distinct, stimulus-locked components, indicating that NE dynamics reflect an integration of global arousal and sensory-driven processes.

• *Supp. Fig 2: Our questions of the buffering effects of nLightG2 were answered in part by the additional in vitro experiments. The in vivo behavioral results, however, are difficult to interpret. In the rebuttal, the authors state that nLightG2 or its control sensor was expressed bilaterally, but in the figure it shows unilateral injection, in line with other experiments in the manuscript. It is unclear that behavior perturbation would be observed if the sensor was delivered unilaterally.*

We would like to thank the Reviewer for the careful attention to detail and apologize for the minor oversight in our rebuttal letter: the mention of bilateral expression was indeed a typographical error. As correctly pointed out by the Reviewer, and as clearly described in Supplementary Figure 2, all behavioral experiments were performed with unilateral sensor expression.

We deliberately chose unilateral expression to best address the technically challenging issue of potential buffering in vivo while closely mirroring a realistic experimental configuration commonly used in neuroscience laboratories. Fiber photometry through a single implanted fiber is a standard and widely adopted approach. Under these expression conditions, our data show no detectable behavioral phenotype that could be attributed to norepinephrine (NE) buffering by the indicator.

Of note, prior work has shown that unilateral artificial perturbation of norepinephrine (NE) release produces robust anxiogenic behavioral effects (see PMID: 28708061). Therefore, if our sensor had buffered NE to a physiologically meaningful extent, we would expect such interference to yield a clear behavioral phenotype in our experiments. We acknowledge that our evidence does not encompass every possible scenario of nLightG2 expression in the brain. For instance, different outcomes might arise if the sensor were expressed broadly or in distinct brain regions. Nonetheless, the absence of a detectable phenotype in our data suggests that any potential buffering by the sensor does not substantially interfere with endogenous NE signaling.

• *The justification for 10Hz low pass filter for a sensor with a much faster on rate is unsatisfactory – very likely, NE peaks would be missed, and the amplitude of change is underestimated. Presumably, the authors already have raw data at higher sampling rate and could apply the appropriate frequency (at least 40Hz given the on rate) for the low pass filter. What do the data look like with such filtering? What is the rationale of not doing this?*

Fiber photometry recordings, particularly when using GPCR-based genetically encoded fluorescent indicators (GEFIs), are inherently prone to various artifacts, including heartbeat-related fluctuations, electrical interference, and other high-frequency (HF) noise sources (see our recent publications addressing these issues, PMID: 38103545, PMID: 37474807). To mitigate such confounds, we applied a series of preprocessing steps, among which temporal filtering plays a critical role. Specifically, the 10 Hz low-pass filter was selected to effectively remove HF noise originating from the animal's heartbeat (~7–12 Hz), electrical components (e.g., 50 Hz), and other sources, without compromising the fidelity of physiologically relevant signals. To demonstrate that this filtering step does not obscure fast norepinephrine (NE) transients, we present in Rebuttal Figure 1 a direct comparison of representative ΔF traces processed with different filter cutoffs and unfiltered data. These comparisons clearly show that all apparent fast fluctuations beyond 10 Hz reflect noise rather than bona fide NE events. Furthermore, the same HF components are also evident in recordings from control animals expressing non-responsive variants (nLightG2-ctr), confirming that these are noise artifacts inherent to the system. Importantly, the 10 Hz low-pass filter efficiently suppresses such noise under baseline conditions in both nLightG2- and nLightR2-expressing animals, resulting in stable, interpretable fluorescence traces without loss of physiological signal content.

Figure Legend:

Rebuttal Figure 1

10Hz Lowpass filtering does not underestimate norepinephrine signals amplitude.

a, Example of nLightG2 photobleaching and motion-corrected signals (ΔF) during cued-fear conditioning without and with 40, 20, 10 and 5Hz lowpass filtering: whole trace (left), and during one representative CS-US pairing (right). Yellow shading indicates the length of tone (CS). Example taken from a mouse in Figure 3 (BLA). **b**, Same as **a** but in a mouse expressing nLightG2-ctr from Supplementary Figure 2 (BLA). **c**, 5s-long example of nLightG2 raw (orange) vs. denoised (blue, 10Hz lowpass filter) signals when no CS or US were presented (baseline). **d**, Same as **c** but for a mouse expressing nLightG2-ctr. Both examples taken from mice in Figure 4 (BLA). **e**, 30s-long example of nLightR2 raw (orange) vs. 50Hz lowpass filter denoised (gray,) vs. 10Hz lowpass filter denoised (blue) before any optogenetic stimulation. Example from Figure 3 (HPC). **f**, Similar to **e**, but in a 60s-long recording during optogenetic stimulation of LC projections.